# SMILE: Sample-to-feature MIxup for Efficient Transfer LEarning

**Xingjian Li**[*]                                                                                     *lixingjian@baidu.com*
*Big Data Lab, Baidu Inc.*
*State Key Lab of IOTSC, University of Macau*

**Haoyi Xiong**[*]                                                                                     *xionghaoyi@baidu.com*
*Big Data Lab, Baidu Inc.*

**Dejing Dou**                                                                                          *doudejing@baidu.com*
*Big Data Lab, Baidu Inc.*

**Chengzhong Xu**                                                                                       *czxu@um.edu.mo*
*State Key Lab of IOTSC, University of Macau*

**Reviewed on OpenReview:** *https://openreview.net/forum?id=czgMCpvrDM*

## Abstract

To improve the performance of deep learning, mixup has been proposed to force the neural networks favoring simple linear behaviors in-between training samples. Performing mixup for transfer learning with pre-trained models however is not that simple, a high capacity pre-trained model with a large fully-connected (FC) layer could easily overfit to the target dataset even with samples-to-labels mixed up. In this work, we propose `SMILE`— Sample-to-feature MIxup for Efficient Transfer LEarning. With mixed images as inputs, `SMILE` regularizes the outputs of CNN feature extractors to learn from the mixed feature vectors of inputs, in addition to the mixed labels. `SMILE` incorporates a mean teacher to provide the surrogate "ground truth" for mixed feature vectors. The sample-to-feature mixup regularizer is imposed both on deep features for the target domain and classifier outputs for the source domain, bounding the linearity in-between samples for target tasks. Extensive experiments have been done to verify the performance improvement made by `SMILE`, in comparisons with a wide spectrum of transfer learning algorithms, including fine-tuning, $L^2$-SP, DELTA, BSS, RIFLE, Co-Tuning and RegSL, even with mixup strategies combined. Ablation studies show that the vanilla sample-to-label mixup strategies could marginally increase the linearity in-between training samples but lack of generalizability, while `SMILE` significantly improves the mixup effects in both label and feature spaces with both training and testing datasets. The empirical observations backup our design intuition and purposes. Our code is available at https://github.com/lixingjian/SMILE.

## 1 Introduction

Performance of deep learning algorithms in real-world applications is often limited by the size of training datasets. Training a deep neural network (DNN) model with a small number of training samples usually leads to the over-fitting issue with poor generalization performance. A common yet effective solution is to train DNN models under transfer learning (Pan et al., 2010) settings using large source datasets. The knowledge transfer

---

[*]Equal Contribution. Correspondence to Haoyi Xiong.

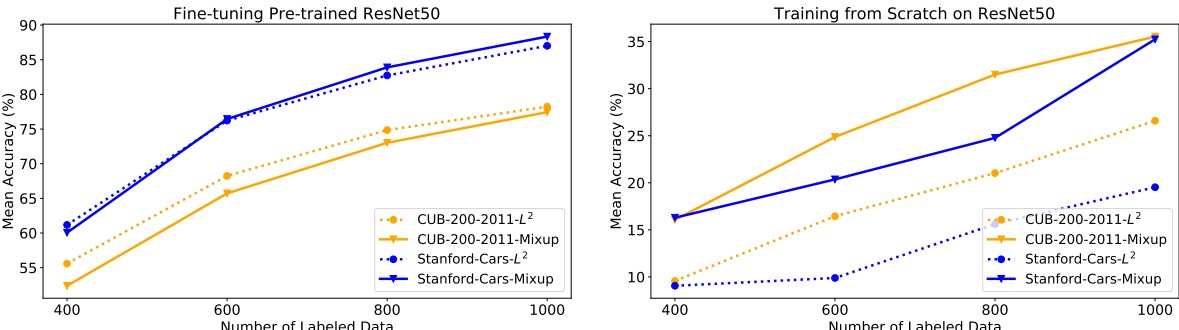

Figure 1: Performance comparison between the $L^2$ regularization and mixup for fine-tuning an ImageNet pre-trained model (left) and training from scratch (right). To simulate scenarios with limited target datasets, we randomly select 50 classes from two transfer learning benchmarks, which are CUB-200-2011 and Stanford-Cars. As seen, Mixup brings remarkable improvements if training from scratch (right), but this does not apply to transfer learning (left).

from the source domain[1] helps DNNs learn better features and acquire higher generalization performance for the pattern recognition in the target domain (Donahue et al., 2014; Yim et al., 2017).

In addition to deep transfer learning, another effective strategy for improve generalization performance of DNN is mixup (Zhang et al., 2018), where the objective is to have DNNs in the learning procedure favor the *linear behaviors* in-between training samples. To achieve the goal, the mixup strategy picks up multiple images from the training set, mixes the samples and labels proportionally to generate a new pair of sample and label for data augmentation. The regularization effects brought by mixup could help control the complexity of DNN models (Hanin & Rolnick, 2019; Vapnik, 2013) while largely improving the robustness and generalization performance (Zhang et al., 2020).

**Research Motivation**   While existing studies about mixup mainly focus on the general setting, to the best of our knowledge, using mixup in transfer learning has rarely been investigated. A straightforward conjecture is that, mixup should be also effective in transfer learning, where typical applications have only limited training examples. In such situations, regularizers aiming to control the model complexity should be beneficial for better generalization. However, the facts are surprisingly just the opposite.   We find that in deep transfer learning, mixup improves the performance with reduced margins or even downgrades the performance when the target dataset is small. See Figure 1 for detailed results.

Considering that either transfer learning and mixup is widely proven beneficial, a natural question is raised as follow,

- *Why mixup tends to negatively affect transfer learning when training samples are limited?*

**Our Analyses**   Two reasons caused the ineffectiveness of mixup in transfer learning. First of all, we find fine-tuning with high capacity pre-trained models CAN overfit to the mixup samples/labels. From mixup, we simply derive a linear interpolation loss to measure the error of linear interpolation between a pair of samples $(x_1, y_1)$ and $(x_2, y_2)$ for the model $f(\cdot)$,

$$\|f(\lambda x_1 + (1 - \lambda)x_2) - (\lambda f(x_1) + (1 - \lambda)f(x_2))\|_2^2 \ , \tag{1}$$

where a lower linear interpolation loss indicates stronger linear behaviors in-between the samples and usually better generalization performance (Zhang et al., 2020). Our experiments however find that fine-tuning with

---

[1]The term *domain* indicates the concept of features or knowledge learned from a task. Please note that, this paper focuses on transferring a pre-trained model on a downstream dataset (labels are available), rather that domain adaptation.

mixup could obtain a low interpolation loss in the training set while suffering a high interpolation loss in the test set ($\geq 25\%$ higher interpolation loss on the testing set than the one on training set, please see also in **Section 5**). This observation indicates that the *linear behaviors* gained by mixup regularization could not well generalize to the testing dataset and overfit to the mixup samples/labels from the training dataset. The second problem is particularly linked to a major challenge in transfer learning called catastrophic forgetting (Li et al., 2018; You et al., 2020). The additional interpolated images generated by mix-up drive the fine-tuned model farther from the starting point, which aggravates the loss of transferable knowledge in the pre-trained model.

Thus, our research intends to study a way to *make mixup strategies generalizable in deep transfer learning settings while alleviating the knowledge loss during fine-tuning.* To achieve the above goal, some non-trivial technical challenges should be tackled.

- *Sample-to-Feature Mixup.* A high-capacity pre-trained model, offering a large quantity of well-trained features, would force a Fully-Connected (FC) Layer to memorize samples and labels mixed-up with trivial updates to weights of its CNN feature extractor. Though some randomized strategies, such as RIFLE (Li et al., 2020), could deepen back-propagation in vanilla transfer learning settings, it is still challenging to reinforce the mixup effects in the CNN feature extractor.

- *Mixed-up Feature Vectors.* To ensure mixup effects in outputs of CNN feature extractors, a possible way is to let CNN feature extractors learn from the mixed-up samples and feature vectors, while the ground-truth feature vectors are usually not available. Thus, surrogate labels of the feature vectors need to be obtained for any sample in the target dataset before having the CNN trained.

- *Cross-Domain Generalizability.* A pre-trained DNN usually is capable of behaving linearly under interpolation of the source dataset. During the fine-tuning procedure, it is reasonable to doubt that such linear behaviors in source domain might be forgotten (Chen et al., 2019). To improve the generalization performance, there thus needs to preserve the linear behaviors in the source domain and transfer such ability to the target domain during fine-tuning.

**Our Work** To address above technical challenges, in this work, we propose **SMILE**—Sample-to-feature Mixup strategies for Efficient Transfer Learning. Instead of regularizing mixup effects in target label spaces (i.e., *sample-to-label* mixup), **SMILE** enables the *sample-to-feature* mixup through regularizing the CNN feature extractor to learn from the surrogate of "fine-tuned" feature vectors of mixed-up images even when the CNN has not yet been well-tuned on the target domain. To the best of our knowledge, this work has made three sets of contributions as follows.

1. We study the problem of regularizing DNNs to enjoy mixup effects under deep transfer learning settings, where the major concern is to avoid the overfitting to mixed-up samples and labels, using a high-capacity pre-trained model but with a small target training dataset. We elaborate the technical issues, and propose to solve the problem through enabling sample-to-feature mixup, where obtaining the feature vectors for mixup and ensuring cross-domain generalizability of linear behaviors become the key challenges.

2. We propose **SMILE**—Sample-to-feature mixup for efficient transfer learning, where sample-to-feature mixup, through either the target deep feature and the source label space, has been used as the core framework of the solution. Given two samples drawn from the target domain as the input, **SMILE** linearly combines two samples proportionally and sends the mixed-up sample to the target model. It constrains the Euclidean distance between the output of target model's CNN feature extractor and a mixed-up feature vector (i.e., linear combination of a mean teacher model's outputs for the two samples) via a *sample-to-feature mixup.* Moreover, to obtain cross-domain generalizability, **SMILE** trains an additional FC classifier for the target network to adapt the target dataset but in the source domain. It regularizes the target network using *sample-to-label* mixup to learn from the linear combination of classification results on source domain, whose labels are also features for the target domain. **SMILE** also optimizes the target model via vanilla sample-to-label mixup using the mixed-up label (i.e., linear combination of ground-truth labels).

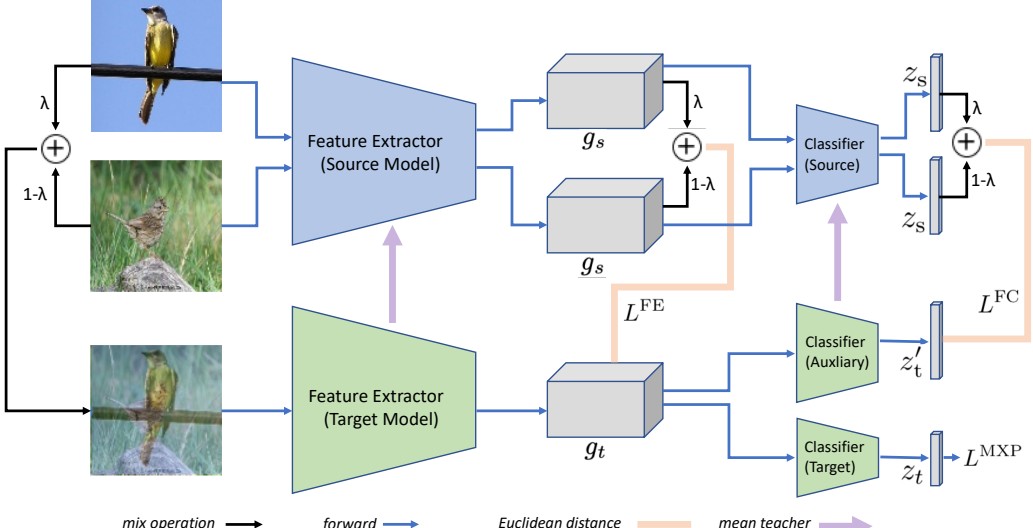

Figure 2: The Architecture of **SMILE**: Deep Transfer Learning with Sample-to-feature Mixup Regularization. To fully exploit the capacity of the source model, we incorporate two components $L^{\text{FE}}$ and $L^{\text{FC}}$, beyond the vanilla mixup $L^{\text{MXP}}$. $L^{\text{FE}}$ focuses on the deep features $g_t$ and $L^{\text{FC}}$ is imposed on the source labels through an auxiliary classifier $z_t'$. We feed original images in the source model, and mixed images in the target model during fine-tuning. Mean teacher is performed on the source model to provide more accurate *pseudo ground truth* for mixed features $g_t$ and $z_t'$. After fine-tuning, we use only the feature extractor and classifier of the target model for prediction.

3. We carry out extensive experiments using a wide range of source and target datasets, and compare the results of **SMILE** with a number of baseline algorithms, including fine-tuning with weight decay ($L^2$) (Donahue et al., 2014), fine-tuning with $L^2$-regularization on the starting point ($L^2$-SP) (Li et al., 2018), DELTA (Li et al., 2019), Batch Singular Shrinkage (BSS) (Chen et al., 2019), RIFLE (Li et al., 2020), Co-Tuning (You et al., 2020) and RegSL (Li & Zhang, 2021) with/without mixup strategies. The experiment results showed that **SMILE** can outperform all these algorithms with significant improvement. The ablation studies show that (1) sample-to-feature mixup design is significantly better than vanilla sample-to-label mixup for deep transfer learning; and (2) the proposed sample-to-label mixup on the source domain can further improve the generalization performance.

## 2 Related Work

In this section, we first introduce the related works from deep transfer learning's perspectives, then we discuss the most relevant work to our study.

### 2.1 Deep Transfer Learning

To enable transfer learning for DNNs, fine-tuning (Donahue et al., 2014) has been proposed to first train a DNN model using the large (and possibly irrelevant) source dataset (e.g. ImageNet), then uses the weights of the pre-trained model as the starting point of optimization and fine-tunes the model using the target dataset. In this way, by sharing the rich and diverse knowledge contained in large source datasets, the fine-tuned model is usually capable of handling the target task with better generalization performance. Furthermore, authors in (Yim et al., 2017; Li et al., 2018; 2019) propose transfer learning algorithms that regularize the training procedure using the pre-trained models, so as to constrain the divergence of the weights and feature maps between the pre-trained and fine-tuned DNN models. Later, the work (Chen et al., 2019; Wan et al., 2019) introduce new algorithms that prevent the regularization from hurting the adaptation to the target

domain in transfer learning, where (Chen et al., 2019) propose to truncate the tail spectrum of the batch of gradients while (Wan et al., 2019) propose to truncate the ill-posed direction of the aggregated gradients. In addition to the aforementioned strategies, algorithms based on the multi-task learning paradigm have been used for deep transfer learning, such as (Ge & Yu, 2017; Cui et al., 2018).

While all above algorithms enable knowledge transfer from source datasets to target tasks, they unfortunately perform poorly due to the catastrophic forgetting and negative transfer. Most transfer learning algorithms (Donahue et al., 2014; Yim et al., 2017; Li et al., 2018; 2019) consist of two steps – pre-training and fine-tuning. Given the features that have been learned in the pre-trained models, either forgetting some good features during the fine-tuning process (*catastrophic forgetting*) (Chen et al., 2019) or preserving the inappropriate features/filters to reject the knowledge from the target domain (*negative transfer*) (Li et al., 2019; Wan et al., 2019) would hurt the performance of transfer learning. In this way, proper compromises should be made between the features learned from both source/target domains during the fine-tuning process, where multi-task learning with Seq-Train (Cui et al., 2018) and Co-Train (Ge & Yu, 2017) might suggest feasible solutions to well-balance the knowledge learned from the source/target domains, through fine-tuning the model with a selected set of auxiliary samples (rather than the whole source dataset) (Cui et al., 2018) or alternatively learning the features from both domains during fine-tuning (Ge & Yu, 2017). A recent study in medical imaging (Wang et al., 2022) employs the Co-Train fashion with auxiliary attributes during both the pre-training and fine-tuning step.

Some other studies (Zhong et al., 2020; Wang et al., 2021; Abuduweili et al., 2021) intend to improve the generalization of the target model by further exploiting the target data. For example, Bi-Tuning (Zhong et al., 2020) incorporates self-supervised learning on top of the standard supervised fine-tuning. Self-tuning (Wang et al., 2021) and Adaptive Consistency Regularization (Abuduweili et al., 2021) consider a more practical scenario that a set of unlabeled target data is available. They find that fine-tuning can be substantially promoted by reasonably utilizing the unlabeled data.

## 2.2 Connections to Our work

The most relevant studies to our algorithm are (Verma et al., 2019; Yun et al., 2019; Li et al., 2019; Chen et al., 2019; Li et al., 2020; You et al., 2020; Wang et al., 2022). While the first two works (Verma et al., 2019; Yun et al., 2019) propose to improve mixup and its derivatives for data augmentation through interpolating the feature spaces, the following three works (Li et al., 2019; Chen et al., 2019; Li et al., 2020; You et al., 2020) focus on improving deep transfer learning through regularizing the feature or label spaces. Authors in (Wang et al., 2022) impose a proximal regularizer to constrain the parameter distance from the pre-trained model, and an auxiliary task that predicts a set of pre-defined attributes such as age, race and so on. In comparison, our method serves as a general algorithm that is free of additional domain knowledge.

The manifold mixup strategy (Verma et al., 2019) has been proposed to smooth the decision boundary of DNN classifiers using mixed-up feature maps and labels, in a *feature-to-label* mixup fashion. On the other hand, CutMix (Yun et al., 2019) also propose a *sample-to-feature* data augmentation strategy, where the algorithm fuses two images into one and forms a new feature map accordingly, with respect to the localizable visual features in two images. Compared to above works, the major technical difficulty of **SMILE** is that above algorithms use feature maps extracted from CNN models directly, while **SMILE** regularizes the output of CNN feature extractor when accurate estimates of feature vectors are not available (the CNN is under fine-tuning to adapt the target dataset).

While (Li et al., 2019; Chen et al., 2019) proposed to improve the feature-wise knowledge distillation or spectral regularization for transfer learning, (Li et al., 2020) studies way to regularize the pre-trained CNN feature extractor, during fine-tuning, by incorporating randomness from FC layers. Compared to above algorithms, **SMILE** is proposed to solve the problem of overfitting to mixup under deep transfer learning settings. Our ablation studies in Section 4 show that the simple combination of fine-tuning and mixup strategies does not perform well in transfer learning settings from both the preservation of linear behaviors and generalization performance aspects. **SMILE** makes unique contributions in proposing novel *sample-to-feature mixup* strategies to improve performance in transfer learning with pre-trained models.

## 3 `SMILE`: **Sample-to-feature Mixup for Efficient Transfer Learning**

In this section, we first introduce the overall framework of `SMILE`, where the architectures of deep transfer learning with Sample-to-feature mixup regularization is presented. Then, we specify the design of the proposed regularizer and discuss the mixup effects incorporated by `SMILE`.

### 3.1 Overall Framework

Given a target training dataset $\mathbf{D} = \{(x_1, y_1), (x_2, y_2)... (x_n, y_n)\}$ and an initial model $\omega_s$ pre-trained with the source dataset, `SMILE` learns a model $\omega$ to adapt the target dataset by a fine-tuning procedure. Since the source model is involved during fine-tuning, we use subscripts $t$ and $s$ to distinguish between the target and source model, superscripts to denote the training iteration. Specifically, `SMILE` initializes the target model $\omega_t^0$ with the pre-trained source model $\omega_s$. For each training iteration, `SMILE` updates the target model $\omega_t$ through minimizing a loss function as follow,

$$\min_{\omega} \left\{ \mathcal{L}(\omega, \omega_s) = \frac{1}{n} \sum_{i=1}^{n} L^{\text{MXP}}(\omega, x_i, y_i) + L^{\text{Reg}}(\omega, \omega_s, x_i) \right\} , \tag{2}$$

where $L^{\text{MXP}}(\omega, x_i, y_i)$ refers to the vanilla sample-to-label mixup loss based on target model on the target domain $\mathbf{D}$, and $L^{\text{Reg}}(\omega, \omega_s, x_i)$ refers to the loss for Sample-to-feature mixup regularization based on both the source and target model. Note that the computation of $L^{\text{Reg}}$ adopts only the target dataset $\mathbf{D}$ as the input and does not rely on labels.

The regularizer $L^{\text{Reg}}$ contains two components, $L^{\text{FE}}$ and $L^{\text{FC}}$, where FE refers to feature extractor and FC refers to fully-connected classifier. For better exploiting the capacity of the source model, $L^{\text{FE}}$ and $L^{\text{FC}}$ leverage the fine-tuned target deep features and the auxiliary source labels, respectively. We present detailed implementations in Section 3.3.

Figure 2 presents the architecture of `SMILE`.

### 3.2 Vanilla Mixup

We first introduce the formulation of vanilla mixup (Zhang et al., 2017). Given a deep neural network $f$, we denote its classifier output as $z$ and deep feature as $g$. Since both $z$ and $g$ depend on the input data and their corresponding parameters, they are formulated as functions for simpleness. Vanilla mixup aims to minimize the linear interpolation loss in the target label space, which can be formulated as

$$L^{\text{MXP}}(\omega) = \mathop{\mathbb{E}}_{\lambda \sim \text{Beta}(\alpha, \alpha)} \mathop{\mathbb{E}}_{x_i, x_j \sim \mathbf{D}} ||z_t \left( \text{Mix}_\lambda(x_i, x_j); \omega \right) - \text{Mix}_\lambda(y_i, y_j)||_2^2 , \tag{3}$$

where the operator $\text{Mix}_\lambda(u, v) = (1 - \lambda) \cdot u + \lambda \cdot v$ refers to the linear combination of two inputs. We follow the vanilla mixup (Zhang et al., 2018) to sample the linear combination coefficient $\lambda$ from a symmetric Beta distribution $\lambda \sim \text{Beta}(\alpha, \alpha)$. The Beta distribution is usually used to sample a random proportion, e.g. $\lambda$ in mixup, since it's the conjugate prior for the Bernoulli distribution. A larger $\alpha$ tends to sample a balanced mixture, i.e. $\lambda$ is more likely near 0.5, and a smaller $\alpha$ leads to $\lambda$ near 0 or 1.

### 3.3 Deep Transfer Learning with Regularization

Omitting the notation of input data $x_i$, we present the Sample-to-feature mixup regularizer as follow,

$$L^{\text{Reg}}(\omega, \omega_s) = \gamma_{\text{FE}} \cdot L^{\text{FE}}(\omega, \omega_s) + \gamma_{\text{FC}} \cdot L^{\text{FC}}(\omega, \omega_s), \tag{4}$$

where $\gamma_{\text{FE}}$ and $\gamma_{\text{FC}}$ refer to the weight of the two terms, the term $L^{\text{FE}}(\omega, \omega_s)$ refers to the sample-to-feature mixup regularizer which borrows general knowledge from the source domain over the target dataset $\mathbf{D}$, and the term $L^{\text{FC}}(\omega, \omega_s)$ refers to the sample-to-label mixup regularizer on the label space of the source domain (e.g., 1000 classes when the model was pre-trained using ImageNet).

Specifically, given a deep neural network $f$, we denote its classifier output as $z$ and deep feature as $g$. Since both $z$ and $g$ depend on the input data and their corresponding parameters, they are formulated as functions for simpleness. Thus, the sample-to-feature mixup regularizer based on target and source models is defined as

$$L^{\text{FE}}(\omega, \omega_s) = \mathop{\mathbb{E}}_{\lambda \sim \text{Beta}(\alpha, \alpha)} \mathop{\mathbb{E}}_{x_i, x_j \sim \mathbf{D}} ||g_t\left(\text{Mix}_\lambda(x_i, x_j); \omega\right) - \text{Mix}_\lambda(g_s(x_i; \omega_s), g_s(x_j; \omega_s))||_2^2 \, , \tag{5}$$

where $g(x_i; \omega)$ refers to the CNN feature extractor output based on weight $\omega$ and the sample $x_i$. This term encourages DNN to learn linear behaviors from samples to hidden features.

Further, the sample-to-label mixup regularizer based on target and source models on the label space of source domain is defined as

$$L^{\text{FC}}(\omega, \omega_s) = \mathop{\mathbb{E}}_{\lambda \sim \text{Beta}(\alpha, \alpha)} \mathop{\mathbb{E}}_{x_i, x_j \sim \mathbf{D}} ||z_{\text{t}}'\left(\text{Mix}_\lambda(x_i, x_j); \omega\right) - \text{Mix}_\lambda(z_{\text{s}}(x_i; \omega_s), z_{\text{s}}(x_j; \omega_s))||_2^2 \, , \tag{6}$$

where $z_{\text{t}}'(x_i; \omega)$ refers to the output of an auxiliary classifier (Fully-Connected) of the target model with $x_i$ on $\omega$ and $z_{\text{s}}(x_i; \omega_s)$ refers to the classifier output of the source model with $x_i$ on $\omega_s$. Both classifiers $z_{\text{t}}'$ and $z_{\text{s}}$ are in the source domain (e.g., with softmax outputs in 1,000 dimensions when the model is pre-trained using ImageNet). More specifically, the FC layer in $z_{\text{t}}'$ is also initialized with the weights of the FC layer in the pre-trained source model $\omega_{\text{s}}$.

The vanilla sample-to-label mixup regularizer $L^{\text{MXP}}$ is derived from the standard implementation of mixup strategy (Zhang et al., 2018) based on target model using the target dataset $\mathbf{D}$. Algorithm 1 presents the design of the overall training procedure of `SMILE`.

### 3.4 Incorporating with a Mean Teacher

Note that an advantage of our method is the compatibility with *mean teacher*, which is widely used in semi-supervised learning problems (Tarvainen & Valpola, 2017) to generate pesudo labels. So far as we know, it has hardly been utilized in deep transfer learning to promote *pesduo features*. Specifically, we periodically update the source model $\omega_s$ by the fine-tuned model in a moving average manner.

---

**Algorithm 1:** Deep Transfer Learning with `SMILE`

---

**Input** : $\mathbf{D}$: target training data, $\omega_{\text{s}}$: pre-trained source model, $\eta$: learning rate, $K$: training iterations, $\alpha$: hyperparameter for Beta distribution ;

**Output :** $\omega_t^K$: final learned target model after $K$ iterations;

**1 begin**

**2**    $\omega_t^0 \leftarrow \omega_{\text{s}}$

**3**    **for** $k \leftarrow 1$ **to** $K$ **do**

**4**      /\*Data Sampling and Mixing\*/

**5**      $\mathbf{B} \leftarrow$ **mini-batch sampling from D**

**6**      $\lambda \sim \text{Beta}(\alpha, \alpha)$

**7**      $\mathbf{B}_m \leftarrow \lambda \cdot \mathbf{B} + (1 - \lambda) \cdot \text{Shuffle}(\mathbf{B})$

**8**      /\*Calculate Vanilla Mixup and Sample-Feature Mixup Loss\*/

**9**      Calculate $L^{\text{MXP}}$ based on $z_t(\mathbf{B}_m, \omega_t^{k-1})$

**10**      Calculate $L^{\text{FE}}$ based on $g_s(\mathbf{B}_m, \omega_s)$ and $g_t(\mathbf{B}, \omega_t^{k-1})$

**11**      Calculate $L^{\text{FC}}$ based on $z_s(\mathbf{B}_m, \omega_s)$ and $z_t'(\mathbf{B}, \omega_t^{k-1})$

**12**      /\*Updating the Target Model with SGD\*/

**13**      $\mathcal{L}(\omega_t^{k-1}, \omega_s) = \gamma_{\text{FE}} \cdot L^{\text{FE}} + \gamma_{\text{FC}} \cdot L^{\text{FC}} + L^{\text{MXP}}$

**14**      $g^k \leftarrow \nabla \mathcal{L}(\omega_t^{k-1}, \omega_s)$

**15**      $\omega_t^k \leftarrow \omega_t^{k-1} - \eta \cdot g^k$

**16**    **return** $\omega_t^K$

---

## 4 Experiments

We evaluate our method on a wide range of tasks, covering different kinds of datasets, pre-trained models, data scales and model architectures. Through exhaustive experiments, **SMILE** is compared against multiple state-of-the-art fine-tuning algorithms including $L^2$ (Donahue et al., 2014), $L^2$-SP (Li et al., 2018), DELTA (Li et al., 2019), BSS (Chen et al., 2019), RIFLE (Li et al., 2020), Co-Tuning (You et al., 2020) and RegSL (Li & Zhang, 2021). To achieve a comprehensive evaluation, we also compare our method with relevant data-augmentation strategies including Mixup (Zhang et al., 2018), Manifold Mixup (Verma et al., 2019) and CutMix (Yun et al., 2019).

Table 1: Characteristics of the target tasks.

| target dataset | task category | source task | architecture | # training | # classes |
|---|---|---|---|---|---|
| CUB-200-2011 | Object Recognition | ImageNet | ResNet-50 | 5,994 | 200 |
| Stanford-Cars | Object Recognition | ImageNet | ResNet-50 | 8,144 | 196 |
| FGVC-Aircraft | Object Recognition | ImageNet | ResNet-50 | 6,677 | 100 |
| MIT-Indoor-67 | Scene Classification | Places365 | ResNet-50 | 5,356 | 76 |
| Food-101 | Object Recognition | ImageNet | EfficientNet-B4 | 75,000 | 101 |

### 4.1 Image Classification

We first present the experiment results based on image classification tasks using a wide range of transfer learning algorithms and datasets.

#### 4.1.1 Datasets and Models

We conduct experiments on three popular object recognition datasets: CUB-200-2011 (Wah et al., 2011), Stanford Cars (Krause et al., 2013) and FGVC-Aircraft (Maji et al., 2013), which are intensively used in state-of-the-art transfer learning literatures (Chen et al., 2019; Li et al., 2020; You et al., 2020). Each of these datasets contains about 6k - 8k training samples. We use ImageNet (Deng et al., 2009) pre-trained ResNet-50 (He et al., 2016) as the source model. For each dataset, we create four subsets with different number of categories and training examples, divided into two experimental groups. For the first group, we first randomly select 25% of all the categories from each of these standard datasets. Then we randomly sample 400 and 800 training samples from the selected categories. For the second group, we use all categories, while evaluate with 15% or 100% training samples respectively, following the practice in existing baselines BSS (Chen et al., 2019) and Co-tuning (You et al., 2020). Among these, the last setting uses the entire dataset, which reflects the performance when adaptation is relatively sufficient, while the remaining three rely more on knowledge transfer from the pre-trained model. The first group simulates real world datasets with relatively fewer categories and more instances pre category, while the second group (15% data) simulates the opposite.

To further confirm the performance improvement by **SMILE** is independent with the choice of pre-trained datasets and model architectures, we conduct additional experiments comparing our method with competitive baselines. Specifically, we use the Places365 (Zhou et al., 2017) pre-trained ResNet-50 to perform fine-tuning on MIT-Indoors-67 (Quattoni & Torralba, 2009), which is a scene classification task. We also evaluate our method on a more powerful model EfficientNet-B4 (Tan & Le, 2019) designed by NAS over a large scale dataset Food-101 (Bossard et al., 2014). The descriptions about the benchmarks used in image classification tasks are summarized in Table 1.

#### 4.1.2 Training Details

We apply standard data augmentation strategies for image pre-processing composed of resizing to $256 \times 256$, random flipping and random cropping to $224 \times 224$ during training. For inference, the test image is resized to $256 \times 256$ and then center cropped to $224 \times 224$. We do not use post-processing methods such as ten-crop

Table 2: Comparison of top-1 accuracy (%) on transfer learning benchmarks. The notation C:X/N:Y refers to using Y examples from X selected categories.

| Dataset | Method | Dataset | | | |
|---|---|---|---|---|---|
| | | C:25%/N:400 | C:25%/N:800 | C:All/N:15% | C:All/N:100% |
| CUB-200-2011 | $L^2$ (Donahue et al., 2014) | 55.59±1.02 | 74.85±0.12 | 44.70±0.17 | 80.64±0.30 |
| | Mixup (Zhang et al., 2017) | 52.39±0.68 | 73.02±0.11 | 44.27±0.31 | 81.86±0.20 |
| | Manifod Mixup (Verma et al., 2019) | 55.38±0.16 | 74.09±0.49 | 49.57±0.30 | 83.09±0.26 |
| | CutMix (Yun et al., 2019) | 28.08±1.26[1] | 59.89±0.94 | 29.73±0.26 | 81.52±0.25 |
| | $L^2$-SP (Li et al., 2018) | 54.38±0.32 | 73.90±0.22 | 45.30±0.23 | 81.58±0.10 |
| | DELTA (Li et al., 2019) | 58.15±0.26 | 75.84±0.08 | 47.88±0.15 | 82.21±0.15 |
| | BSS (Chen et al., 2019) | 54.99±0.73 | 74.14±0.34 | 46.41±0.09 | 81.10±0.04 |
| | RIFLE (Li et al., 2020) | 53.68±0.89 | 73.05±1.09 | 44.13±0.38 | 81.94±0.06 |
| | Co-Tuning (You et al., 2020) | 57.98±0.08 | 75.11±0.47 | 49.98±0.23 | 82.60±0.03 |
| | RegSL (Li & Zhang, 2021) | 57.62±0.88 | 75.51±0.44 | 46.92±0.28 | 80.20±0.17 |
| | **SMILE** | **62.13±0.55** | **77.27±0.35** | **51.73±0.04** | **83.62±0.07** |
| Stanford-Cars | $L^2$ (Donahue et al., 2014) | 61.17±0.36 | 82.73±0.59 | 43.01±0.53 | 90.14±0.12 |
| | Mixup (Zhang et al., 2017) | 60.25±0.68 | 83.60±0.02 | 45.73±0.15 | 91.51±0.18 |
| | Manifod Mixup (Verma et al., 2019) | 64.38±0.73 | 85.01±0.18 | 50.53±0.22 | 91.88±0.16 |
| | CutMix (Yun et al., 2019) | 47.89±0.78 | 76.71±0.36 | 37.62±0.14 | **92.56±0.20** |
| | $L^2$-SP (Li et al., 2018) | 61.00±0.28 | 82.05±0.05 | 44.12±0.33 | 90.61±0.12 |
| | DELTA (Li et al., 2019) | 62.05±0.13 | 82.1±0.44 | 43.27±0.27 | 90.86±0.08 |
| | BSS (Chen et al., 2019) | 64.97±0.69 | 83.81±0.39 | 47.45±0.23 | 91.14±0.04 |
| | RIFLE (Li et al., 2020) | 62.85±0.22 | 83.57±0.43 | 43.61±0.07 | 91.08±0.12 |
| | Co-Tuning (You et al., 2020) | **66.05±0.41** | 81.05±0.39 | 44.29±0.42 | 91.19±0.11 |
| | RegSL (Li & Zhang, 2021) | 60.12±0.63 | 82.91±0.08 | 42.52±0.37 | 91.02±0.05 |
| | **SMILE** | 65.17±1.11 | **85.90±0.16** | **50.93±0.17** | 92.21±0.05 |
| FGVC-Aircraft | $L^2$ (Donahue et al., 2014) | 59.63±1.11 | 79.57±0.18 | 51.13±0.45 | 88.27±0.51 |
| | Mixup (Zhang et al., 2017) | 65.20±0.80 | **84.53±0.62** | 54.42±0.55 | 89.33±0.17 |
| | Manifod Mixup (Verma et al., 2019) | 61.10±0.56 | 80.40±1.90 | 57.97±0.61 | 89.53±0.24 |
| | CutMix (Yun et al., 2019) | 53.50±0.80 | 77.60±0.73 | 44.10±0.78 | 88.48±0.27 |
| | $L^2$-SP (Li et al., 2018) | 54.70±0.73 | 76.13±0.82 | 48.85±0.70 | 87.97±0.66 |
| | DELTA (Li et al., 2019) | 53.47±0.24 | 71.73±1.02 | 51.05±0.38 | 88.92±0.25 |
| | BSS (Chen et al., 2019) | 61.40±1.13 | 81.47±0.24 | 52.61±0.11 | 88.47±0.16 |
| | RIFLE (Li et al., 2020) | 60.97±0.49 | 79.87±0.38 | 52.13±0.31 | 89.45±0.44 |
| | Co-Tuning (You et al., 2020) | 62.98±0.72 | 80.03±0.04 | 52.05±0.43 | 88.19±0.33 |
| | RegSL (Li & Zhang, 2021) | 61.87±0.37 | 79.40±0.92 | 51.64±0.43 | 88.87±0.26 |
| | **SMILE** | **68.40±0.33** | **84.57±0.29** | **60.04±0.33** | **90.16±0.15** |

ensemble (Liang et al., 2020). We train all models using SGD with the momentum of 0.9, weight decay of $10^{-4}$ and batch size of 48. We train 15,000 iterations for Food-101 considering its large scale and 9,000 iterations for the remaining datasets. The initial learning rate is set to 0.001 for MIT-Indoor-67 due to its high similarity with the pre-trained dataset Places365 and 0.01 for the remaining. The learning rate is divided by 10 after two-thirds of total iterations. Each experiment is repeated five times and we report the average top-1 classification accuracy and standard derivations for uncertainty quantification.

For hyper-parameter search, we use a simple three-fold cross validation on the original training set from $\gamma_{FE} \in [0.01, 0.1]$ and $\gamma_{FC} \in [0.01, 0.1]$. The selected best configurations are used in all experiments. As for baseline methods, we use the recommended choices of hyper-parameters reported in corresponding papers.

---

[1]We notice that CutMix performs surprisingly worse in low-data regime (e.g., less than 1000 training examples in total). However, when fine-tuning with full data, CutMix always outperforms vanilla fine-tuning. This phenomenon is consistent with our preliminary experiments in Introduction. A conjecture is that, when training data are insufficient, the operation of cutting and replacing patches might cause much severer over-fitting risks. The CutMix results can be reproduced by our published code.

Table 3: Comparison of top-1 accuracy (%) with different transfer learning algorithms on more task types and architectures.

| Dataset | Method | Sampling Rates | | |
|---|---|---|---|---|
| | | 30% | 50% | 100% |
| MIT-Indoor-67 | $L^2$ (Donahue et al., 2014) | 78.68±0.20 | 80.80±0.18 | 82.00±0.21 |
| | Mixup (Zhang et al., 2017) | 77.44±0.44 | 80.28±0.28 | 82.87±0.50 |
| | DELTA (Li et al., 2019) | 80.80±0.22 | 82.80±0.25 | 83.67±0.18 |
| | BSS (Chen et al., 2019) | 78.23±0.50 | 80.35±0.28 | 82.15±0.22 |
| | RIFLE (Li et al., 2020) | 76.76±0.08 | 78.71±0.33 | 81.78±0.07 |
| | SMILE | **82.00±0.14** | **83.54±0.20** | **85.37±0.16** |
| Food-101 | $L^2$ (Donahue et al., 2014) | 80.25±0.28 | 83.43±0.15 | 86.77±0.03 |
| | Mixup (Zhang et al., 2017) | 82.63±0.11 | 84.93±0.06 | 87.82±0.06 |
| | DELTA (Li et al., 2019) | 81.38±0.08 | 84.07±0.06 | 87.34±0.07 |
| | BSS (Chen et al., 2019) | 81.13±0.04 | 83.96±0.09 | 87.33±0.03 |
| | RIFLE (Li et al., 2020) | 81.13±0.04 | 83.82±0.02 | 87.29±0.11 |
| | SMILE | **82.84±0.16** | **85.25±0.09** | **88.20±0.10** |

### 4.1.3 Results

As observed in Table 2, our proposed `SMILE` achieves remarkable improvements to vanilla fine-tuning on three standard benchmarks, and outperforms all state-of-the-art methods. As the size of training data becomes smaller, our method yields more significant benefits, e.g. `SMILE` outperforms vanilla fine-tuning by more than 8% on FGVC-Aircraft when only 15% training samples are used.

Our method is shown scalable to more challenging datasets. For MIT-Indoor-67, vanilla fine-tuning with a small learning rate is quite competitive as the pre-trained model is highly adaptable for the target task. While for large-scale dataset Food-101, the benefit from all fine-tuning algorithms becomes less. As shown in Table 3, `SMILE` still delivers decent performance on these datasets.

As for the time complexity, although `SMILE` requires an extra forward pass, the actual running time increases less than 20% against vanilla fine-tuning.

### 4.2 Natural Language Processing

We also evaluate `SMILE` on the text classification task using powerful transformer-based architecture, showing that our method can be applied to NLP tasks.

### 4.2.1 Datasets and Models

To carry out experiments of transfer learning for NLP tasks, we use the fine-grained sentiment classification task SST-5, which offers the Stanford Sentiment Treebank datasets with five categories. The pre-trained model in this experiment is base model of BERT (Devlin et al., 2018) with 12 transformer blocks and 12 attention heads.

### 4.2.2 Training Details

We fine-tune the pre-trained BERT model with the batch size to 24 for 3 epochs, using Adam optimizer with a learning rate of $2 \times 10^{-5}$ while incorporating with deep transfer learning algorithms including $L^2$-SP and BSS (Chen et al., 2019) and vanilla Mixup.

### 4.2.3 Results

Results are shown in Table 4, where we include performance based on LSTM (Tai et al., 2015), CNN (Kim, 2014), and the vanilla BERT$_{base}$ reported in (Munikar et al., 2019) for comparisons. We also find that both

mixup and `SMILE` outperform standard fine-tuning and `SMILE` achieves more improvements. Regularizers L2-SP and BSS without mixup are not superior to standard fine-tuning in this task.

## 5 Analysis

### 5.1 Ablation Study

We here present an ablation study to exhibit the unique contribution corresponding to each component in our framework. Specifically, we evaluate the performances of `SMILE` by removing $L^{\mathrm{FE}}$ and $L^{\mathrm{FC}}$ respectively. As observed in Table 5, while they both make non-trivial contributions, the influence of the sample-to-feature mixup regularizer $L^{\mathrm{FE}}$ is more significant than the feature-to-label regularizer $L^{\mathrm{FC}}$.

Furthermore, we consider simple combinations of mixup strategies and state-of-the-art transfer learning algorithms. Specifically, we employ DELTA (Li et al., 2019) to improve knowledge distillation and RIFLE (Li et al., 2020), BSS (Chen et al., 2019) to alleviate negative transfer, both on top of the vanilla mixup. As shown in Figure 3, though such a straightforward combination with vanilla mixup sometimes improves either transfer learning and mixup, `SMILE` is still significantly superior.

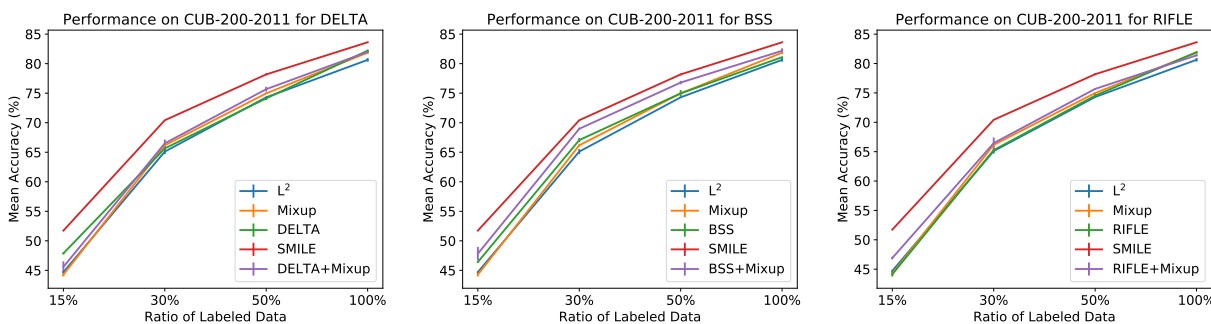

Figure 3: Comparison of top-1 accuracy (%) with various SOTA transfer learning baselines combined with Mixup.

### 5.2 Discussions on Linear Interpolation Effects

One major assumption of our work is that *fine-tuning with pre-trained models would overfit to mixed-up samples and labels*. In order to verify this assumption and interpret the performance improvement of `SMILE`, we now investigate the linear interpolation effects in fine-tuning. It is worth noting that, the terminology *overfit* and *generalize* in this section are particularly w.r.t. the linear interpolation, rather than the model accuracy in general sense. For example, *a model overfits to mixed-up samples and labels* means, the model remembers all interpolated labels for the mixed training samples, but a mixture of two test samples fails to get the prediction that lies on the linear interpolation of their respective predictions.

Table 4: Experimental results on NLP task SST-5.

| Methods | Accuracy |
|---|---|
| LSTM (Tai et al., 2015) | 46.4 |
| CNN (Kim, 2014) | 48.0 |
| BERT$_{base}$ (Devlin et al., 2018) | 53.2 |
| BERT$_{base}$ w. Mixup | 53.7 |
| BERT$_{base}$ w. L$^2$-SP (Li et al., 2018) | 53.2 |
| BERT$_{base}$ w. BSS (Chen et al., 2019) | 53.4 |
| BERT$_{base}$ w. `SMILE` | **54.6** |

Table 5: Ablation Study on the CUB-200-2011 dataset.

| Methods | Sampling Rates | |
|---|---|---|
| | 30% | 100% |
| `SMILE` | 70.42 | 83.62 |
| `SMILE` w/o. $L^{\mathrm{FC}}$ | 69.68±0.12 | 83.11±0.19 |
| `SMILE` w/o. $L^{\mathrm{FE}}$ | 68.15±0.27 | 82.92±0.20 |

### 5.2.1 Measuring Linear Interpolation Effects

To quantify the linear interpolation effects, we define metric to quantify the effect of linear interpolation. Derived from standard mixup (Zhang et al., 2018), we introduce a generalized form of interpolation loss (IL) w.r.t a function $f$ employing its own outputs as labels, eliminating the influence of the faithfulness of the approximation, i.e. how the learned function fits the ground truth $f^*$, as follows:

$$
\begin{aligned}
\text{IL}(f) =& \mathbb{E}_{x,x'\sim D}\mathbb{E}_{\delta_1,\delta_2\sim P_\delta}\mathbb{E}_{\lambda\sim P_\lambda} D^{it}_\lambda(f(\text{Mix}_{\lambda\delta_1+(1-\lambda)\delta_2}(x,x\prime)),\\
& f(\text{Mix}_{\delta_1}(x,x\prime)), f(\text{Mix}_{\delta_2}(x,x\prime))),
\end{aligned}
\tag{7}
$$

where $D^{it}_\lambda$ refers to the Euclidean distance between the output w.r.t the interpolated inputs and the proportionally mixed outputs. $\lambda$ conforms to the Beta distribution as described in Section 3.2, and $\delta_1, \delta_2$ are sampled from a uniform distribution between 0 and 1. Note that Equation (7) is a metric for evaluating to what degree does a model favor linear behaviors, rather than an optimization objective.

Compared with original form of linear interpolation loss used in standard mixup (Zhang et al., 2018), the metric defined in Equation (7) has the following two merits.

- Equation (7) is feasible for measuring linear interpolation effects for both the feature layer (noted as Feature-IL when considering the CNN feature extractor as the function $f$) and the label outputs (noted as Label-IL when considering the classifier's outputs as $f$).

- Equation (7) relies on the network's own outputs rather than external labels, and thus the influence of label fitting (e.g. model accuracy) is disentangled from the evaluation of linear interpolation.

### 5.2.2 Sample-to-Feature Mixup: Linear Interpolation Effects and Generalization

We use Eq 7 to measure Label-IL using the classifier outputs and Feature-IL using the last hidden layer of ResNet-50 for different transfer learning methods, with CUB-200-2011 (with 30% sampling rates) as the training set for all methods. Several arguments can be deduced from results in Table 6.

- *More Data, Better Generalization, and Lower Label-IL and Feature-IL.* There is no doubt to assume that, in practice, a model trained with more data should enjoy better generalization performance. In addition to improve the testing accuracy , we find that, when we involve additional training samples, both Label-IL and Feature-IL would be lower on the testing sets, compared to vanilla fine-tuning.

- *Fine-tuning with vanilla mixup is NOT generalizable even in the label space, due to the lack of linear interpolation in feature spaces.* As shown in Table 6, although Label-IL of the vanilla mixup is significantly lower on the training set than other methods, its Label-IL is high on the testing set (not generalizable). Furthermore, compared to other methods on both training/testing sets (even Fine-tuning on the testing set), Feature-IL of the vanilla mixup is high, i.e., poor linear interpolation in feature spaces.

- *Sample-to-Feature Mixup could ensure the generalizability of mixup effects in the label space, as* `SMILE` *is with low Feature-IL and Label-IL on both training and testing sets.* While `SMILE` achieves the lowest Feature-IL on both training and testing datasets, it also achieves the lowest testing Label-IL. The comparisons with vanilla mixup suggest that doing mixup in the label space is just not enough for fine-tuning. Besides the quantitative results, we in Appendix B present some visualized cases of interpolation behaviors on the feature space with different fine-tuning methods.

These arguments solidify our motivation of sample-to-feature mixup for fine-tuning.[2]

---

[2]Note that the over-fitting in the label space may not be directly calibrated with training/test accuracy as there exists other factors influence the accuracy, e.g. mixup also benefits from the effect of label smoothing (Singh & Bay, 2019).

Table 6: Feature-IL and Label-IL for different fine-tuning methods over the training (sampling the CUB-200-2011 training set by 30%) and testing dataset. Lower is better. Add. Data refers to involving the remaining 70% training examples for fine-tuning. However, the interpolation loss for the training set is still calculated on the original 30%.

| Method | Label-IL | | Feature-IL | |
|---|---|---|---|---|
| | Train | Test | Train | Test |
| Finetune | 1.80 | 1.92 | 1.92 | 1.93 |
| Finetune + Add. Data | 1.85 | 1.88 | 1.58 | 1.63 |
| Finetune + MXP | **1.65** | 2.00 | 1.98 | 2.02 |
| **SMILE** | 1.75 | **1.82** | **1.48** | **1.53** |

### 5.2.3  Feature-to-Label Mixup: déjà vu can help.

To enforce linear behaviors on features, **SMILE** inherits the Feature-to-Label classifier (i.e., the FC layer) from the source model as the initialization of fine-tuning. It is because we assume that the label space of the source task is partially overlapped with the target task. Thus, the FC layer with a considerable number of parameters contains useful information for the target task. This has been investigated by correlating the label spaces between these two tasks in recent studies (You et al., 2020). Furthermore, although it is impossible for **SMILE** to mix the feature vectors extracted from the fine-tuned CNN during the fine-tuning procedure, Feature-to-Label Mixup still works well as the FC classifier has been well-trained on the source datasets, which provides rich semantic information.

### 5.3  Discussions on Catastrophic Forgetting

Here we present discussions to confirm our hypothesis stated in the introduction, i.e. the vanilla mixup aggravates the risk of catastrophic forgetting, while our approach alleviates this problem by reusing rich source knowledge. As suggested by existing literature (Li et al., 2018; Gouk et al., 2021; Li & Zhang, 2021), we use the parameter distance between the fine-tuned and pre-trained model to measure the degree of catastrophic forgetting. We use the same experimental setting as that used in Section 5.2. As shown in Table 7, both mixup and **SMILE** get the parameter distance even larger than vanilla fine-tuning with double training examples, while **SMILE** alleviates the deviation caused by mixup.

Table 7: Parameter distance between the fine-tuned and pre-trained model with different methods. The distance is calculated as the summation of the distance w.r.t. each layer, measured by the Euclidean distance between two tensors. The FC layers are not included when calculating the distance.

| Method | Finetune | Finetune (2x data) | Finetune+mixup | Finetune+**SMILE** |
|---|---|---|---|---|
| Distance | 32.3 | 37.3 | 90.1 | 60.7 |

### 5.4  Role of Source Model

Our method is effective particularly in transfer learning, and a reliable teacher model is vital to the proposed framework. In the setting of fine-tuning, the source model acts as a good starting point to provide supervision for both cross-domain mixed labels and in-domain mixed features. Thereby, our method is not directly feasible for general-purpose supervised learning. The reason are two folds.

In general supervised learning, i.e., learning a single task from scratch, the cross-domain sample-to-label regularier $L^{FC}$ is no longer applicable, since no auxiliary tasks are available. This makes our framework incompatible with general supervised learning.

The source model also plays an essential role for supervision of mixed features. If the teacher is not trustworthy enough, the effects of sample-to-feature mixup cannot be guaranteed. We design two groups of experiments to

verify this. In the first group, we evaluate **SMILE** with degenerated teachers, including one without knowledge preserving from the source model, and the other without adaptation to the target data. They are denoted by "w/o Source" and "w/o Target" respectively. The second group simulates general supervised learning, where only original mixup and sample-to-feature mixup are involved in **SMILE**. Specifically, results in Table 8 show that preserving the source weights contributes more than adapting to the target data, in terms of sample-to-feature mixup. Experiments on the second group further backup the analysis. As shown in Table 9, when training from scratch, **SMILE**is inferior to vanilla Mixup, indicating that the sample-to-feature component has negative influence if using a low-quality teacher.

Table 8: Evaluation of fine-tuning on CUB-200-2011 C:25%/N:800.

| Mixup | SMILE | SMILE w/o Target | SMILE w/o Source |
|-------|-------|------------------|------------------|
| 73.02 | 77.27 | 75.72 | 73.45 |

Table 9: Evaluation of training from scratch on CUB-200-2011 C:25%/N:800.

| $L^2$ | Mixup | SMILE |
|-------|-------|-------|
| 16.97 | 25.36 | 21.86 |

## 6 Conclusion

In this work, we figure out the difficulty of applying mixup to transfer learning, and introduce **SMILE**—Sample-to-feature Mixup strategies for Efficient Transfer Learning. Beyond a direct combination of fine-tuning and mixup, **SMILE** pursues generalizable linear behaviors through incorporating both features of the target domain and the label space of the source domain. We conduct extensive experiments using a wide spectrum of target datasets. Results show that **SMILE** can significantly promote the effectiveness of fine-tuning and outperform various competitive fine-tuning algorithms. Ablation studies and empirical discussions further backup our design intuition and purposes.

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

## A   Detailed Results for Figure 1

Here we provide detailed results for Figure 2, with the mean accuracy and standard deviation among five random trials for each experiment. We additionally report performance for 200 training samples for clearer comparison. Results in Table 10 clearly show that, in transfer learning, Mixup performs worse (compared against the baseline) as fewer training samples are available. However, when training from scratch, Mixup consistently improves the performance with a large margin.

Table 10: Accuracies with standard deviations corresponding to Figure 2. TL refers to transfer learning, where we fine-tuning an ImageNet pre-trained checkpoint to adapt the new task. ST refers to standard training, i.e. training from a random initialization.

| Dataset | Method | Size of Training Set | | | | |
|---------|--------|------|------|------|------|------|
| | | 200 | 400 | 600 | 800 | 1000 |
| CUB(TL) | Baseline | 32.98±0.13 | 55.59±1.02 | 68.24±0.35 | 74.85±0.12 | 78.83±0.42 |
| | Mixup | 27.25±0.18(-5.73) | 52.39±0.68(-3.20) | 65.72±0.58(-2.52) | 73.02±0.11(-1.83) | 77.44±0.14(-1.39) |
| Cars(TL) | Baseline | 38.13±0.60 | 61.17±0.36 | 76.22±0.44 | 82.73±0.59 | 87.01±0.07 |
| | Mixup | 33.85±0.22(-4.28) | 60.25±0.68(-0.92) | 77.45±0.57(+1.23) | 83.60±0.02(+0.87) | 88.34±0.22(+1.33) |
| CUB(ST) | Baseline | - | 9.57±0.49 | 16.45±0.21 | 19.06±0.62 | 26.11±3.24 |
| | Mixup | - | 16.07±1.20(+6.50) | 24.87±0.80(+8.42) | 32.49±1.22(+13.43) | 35.53±0.96(+9.42) |
| Cars(ST) | Baseline | - | 9.06±0.90 | 9.89±2.80 | 15.61±1.15 | 19.53±0.68 |
| | Mixup | - | 16.29±2.22(+7.23) | 20.36±1.92(+10.47) | 24.77±0.41(+9.16) | 35.24±0.69(+15.71) |

## B  Demonstrated Effects of Feature Interpolation

Here we present show cases for comparison in feature interpolation among different algorithms. To obtain the interpolation points, we first randomly select a pair of images and then generate five mixed inputs with the interpolation coefficients $\lambda$ of [0.6, 0.7, 0.8, 0.9, 1] respectively. Forward computation is performed given these mixed inputs and then, their corresponding deep features are extracted and projected to the 2-D space using PCA. We plot the interval of $\lambda > 0.5$ for better demonstration of projection as the same sample dominates the interpolated result when all $\lambda$ lie in either $(0, 0.5)$ or $(0.5, 1)$. Results of four random pairs are illustrated in Figure 4.

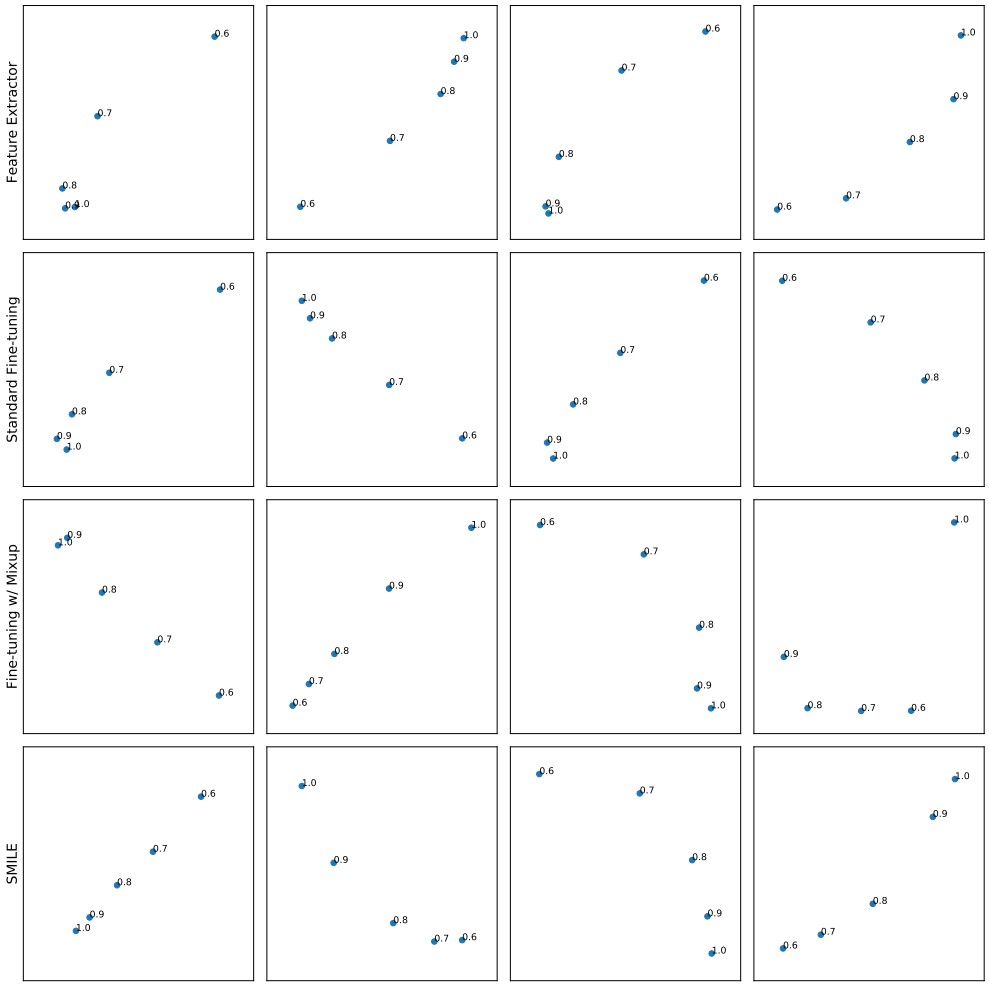

Figure 4: Visualizations of feature interpolation behaviors for different fine-tuning methods. We extract the representations from the last hidden layer which are 2048 dimensional feature vectors and then project them into the 2-D space using PCA. Each column corresponds to the projected deep features generated by putting forward the interpolation of a random pair of images. The number marked next to the point refers to the interpolation coefficient $\lambda$.

