# OpenReview forum: "SMILE: Sample-to-feature Mixup for Efficient Transfer Learning"
_TMLR — Accepted by TMLR_

### Review · Reviewer_aFiM · 2022-09-27

**Summary Of Contributions:**

The authors found that mixup data augmentation technique doesn’t help much in a transfer learning setup especially when the target dataset is small. They therefore propose to regularize both sample-to-feature mixup and sample-to-label mixup by leveraging a mean teacher feature extractor and a classifier to boost the model’s performance on the target task. They perform experiments on several datasets for image classification and a natural language processing task. The proposed method outperformed a list of baseline methods.

**Broader Impact Concerns:**

No ethical concerns observed.

**Requested Changes:**

- Justify on the motivation of the study;
- Discuss the relationship of a similar work in the literature;
- Improve the language and flow of the methodology section;
- Theoretical analysis of the proposed method;
- Add cross-domain generalization experiments;


**Strengths And Weaknesses:**

Strengths:
- The authors performed a lot of experiments on many datasets and tasks;
- The proposed method shows substantial improvement compared to other baselines;

Weaknesses:
- The study is motivated by the observation that mixup doesn't help when transferring models to a small target dataset compared to training from scratch. But please note that the performance of a transferred model and the scratch model is at a very different scale. The authors also warn that mixup in transfer learning might be harmful. However, the difference of the performance between a pain transferred model and a mixup transferred model might be not statistically significant;
- The proposed strategy is very similar to the proxy regularizer introduced in [1] but in a mixup setting. Perhaps the authors want to discuss the connection;
- The methodology section is very hard to follow in general. The authors need to fix the logic and flow. For example, $L^{FE}$ and $L^{FC}$ are not defined in section 3.1;
- Is there any theoretical justification of the proposed regularization method?
- The caption of Figure 2 is not self-contained. Please elaborate on the design of the proposed architecture;
- Where are the cross-domain (out-of-sample) generalization experiments mentioned in the introduction?

[1] Wang, R., Chaudhari, P. and Davatzikos, C., "Embracing the disharmony in medical imaging: a simple and effective framework for domain adaptation." Medical Image Analysis 76 (2022): 102309.

---

> ### Author Response · Authors · 2022-10-31
> **Response to Reviewer aFiM**
>
> Thank you for your critical comments and useful suggestions. Based on your advices, we have revised the manuscript. Here we list the weaknesses you figured out and present our responses point-by-point.
>
> **W1. The study is motivated by the observation that mixup doesn't help when transferring models to a small target dataset compared to training from scratch. But please note that the performance of a transferred model and the scratch model is at a very different scale. The authors also warn that mixup in transfer learning might be harmful. However, the difference of the performance between a pain transferred model and a mixup transferred model might be not statistically significant;**
>
> We agree that the performance of training from scratch and that of transfer learning differ in a great scale.  Our comparison with training from scratch intends to indicate that, the risk of using mixup in fine-tuning is caused by not only the small data size, but also the pre-trained models.
>
> In terms of the difference between the plain transferred and mixup transferred model, we confirm the results are statistically significant by detailed accuracy numbers with their standard deviations. We add the corresponding table in Appendix A. These results show that, when fewer training samples are available, the mean accuracy decrease caused by mixup is obviously larger than the standard deviation.
>
> **W2. The proposed strategy is very similar to the proxy regularizer introduced in [1] but in a mixup setting. Perhaps the authors want to discuss the connection;**
>
> We add discussion about the proximal regularizer introduced in [1]. Specifically, we briefly introduce [1] in the second paragraph of Section 2.1, and discuss the connection with our work in the first paragraph of Section 2.2.
>
> *[1] Wang, R., Chaudhari, P. and Davatzikos, C., "Embracing the disharmony in medical imaging: a simple and e ective framework for domain adaptation." Medical Image Analysis 76 (2022): 102309.*
>
> **W3. The methodology section is very hard to follow in general. The authors need to fix the logic and flow. For example, $L^{FE}$ and $L^{FC}$ are not defined in section 3.1;**
>
> We add an introduction about $L^{FE}$ and $L^{FC}$, and refer readers to Section 3.3 for detailed formulations and explanations. Besides, we have fixed notation errors in the methodology section.
>
> **W4. Is there any theoretical justification of the proposed regularization method?**
>
> Our work is mainly an empirical algorithm, which is motivated by counter-intuitive observations that mixup tends to harm fine-tuning as training samples reduce.  In general, our work follows the same assumption that linear behaviors (if generalizable) can improve the generalization performance of deep models. Our work figures out the risk of over-fitted linear behaviors in transfer learning scenarios, and proposes practical solutions.
>
> **W5. The caption of Figure 2 is not self-contained. Please elaborate on the design of the proposed architecture;**
>
> We add detailed explanations in the caption of Figure 2.
>
> **W6. Where are the cross-domain (out-of-sample) generalization experiments mentioned in the introduction?**
>
> We are sorry for the misleading, but we did not claim the capacity of out-of-sample (or out-of-domain) generalization. The expression “cross-domain generalizability” here means we intend to preserve the linear behaviors achieved by the source domain, and transfer the ability to the target domain. In other words, our method aims to improve the in-domain (within the target data distribution) generalization, through properly exploiting the capacity of cross-domain data (the source domain).

---

### Review · Reviewer_Dv9f · 2022-10-04

**Summary Of Contributions:**

This paper focuses on the problem of transfer learning, the authors try to answer how to fine-tune pre-trained models to downstream tasks. Specifically, they find that mix-up overfits when fine-tuning deep neural networks. To solve this problem, they propose a mix-up based fine-tuning method, SMILE, to enable a better generalization in the downstream tasks.
Specific technical details: SMILE applies mix-up to both the input space and the output space of the training model (feature vectors and classifier outputs), and performs regularization to solve the problem of catastrophic forgetting. In the experiments, the paper compares SMILE with SOTA methods in the fine-tuning literature using image classification and NLP tasks, and shows that SMILE is effective in enhancing the generalization ability of deep models.

**Broader Impact Concerns:**

No.

**Requested Changes:**

Though good experimental results are provided, this paper should add more insights to make the proposed method convincing:
1. Why mix-up can benefit transfer learning?
2. Why SMILE can solve the essential problem of transfer learning?
3. Why mix-up overfits in transfer learning?
4. For further improving the paper, The authors should reorganize the research question and give some correspondences between research questions and experimental results.

**Strengths And Weaknesses:**

Strengths:
1. This paper shows the effectiveness of the proposed SMILE in a wide range of datasets. It is convincing to me that SMILE is useful in fine-tuning.
Weaknesses:
1. Though good experimental results are shown in a wide range of datasets, it is still not clear why SMILE is a good solution. The whole story is based on the assumption that mix-up can benefit transfer learning. However, no explorations about the relationship between mix-up and transfer learning are shown. To further improve the paper, I suggest the authors explore what is the key factors in transfer learning and why mix-up can benefit the transfer learning process.
2. This paper lacks insights. I think it should be some correspondences between research questions and experimental results. A good paper should solve the targeted problem and tell readers why the proposed method can solve the principal difficulty. There should be more clarification about why SMILE can solve the essential problem of transfer learning.
3. Vanilla mix-up overfits in transfer learning, this is why SMILE is proposed. Insights about why mix-up overfits should be explored.
4. Considering the above three weaknesses, it should be the case that SMILE is a combination of existing technologies and no extra insights are provided.

---

> ### Author Response · Authors · 2022-10-31
> **Response to Reviewer Dv9f**
>
> Thank you for your critical comments and useful suggestions. We understand that your major concerns are the motivation and insight of our paper. These concerns can be summarized by the following three questions in a logical order, to which we respond one-by-one and make corresponding improvements in our revision.
>
> **Q1. Why can mix-up can benefit transfer learning? More accurately, why we expect mix-up to benefit transfer learning (though actually not).**
>
> **Q2. Why mix-up overfits in transfer learning?**
>
> **Q3. Why SMILE solves the problem where mix-up fails?**
>
> **A1.** Mixup is a general learning strategy widely used in deep learning. While existing studies on mixup mainly focus on supervised learning in general settings, to the best of our knowledge, using mixup in transfer learning (particularly the fine-tuning paradigm in our work) has rarely been investigated. A straightforward conjecture is that, mixup in transfer learning should be as effective as the one in supervised learning, where typical applications have only limited training examples. Such an expectation sounds reasonable since mixup can be regarded as a kind of regularization that enforces the learned model simpler (behaving linearly). This is the initial motivation of our work, whereas we surprisingly find that the facts are just the opposite. Our preliminary experiments in Figure 1 show that, mixup performs worse (compared to its naive fine-tuning counterpart) as the number of training examples decreases. According to our analyses, the failure of mixup lies in overfitting.
>
> **A2.** “Why mixup overfits in transfer learning” is actually a somewhat complex but interesting question. We did not discuss it much in details as we did not regard it as the main technical contribution of our work. Following your advice, we add these analyses into the revised manuscript, and here present a summarization of key arguments.
>
> From a perspective of statistical learning, the risk of overfitting undoubtedly increases when training examples are limited. However, this general principle alone fails to explain why fine-tuning with mixup performs relatively worse compared to vanilla fine-tuning, since they face the same challenge of insufficient supervision when reducing training examples.
>
> The deep reason is from the following two folds. The first explains why mixup overfits when training examples are insufficient. The section further explains why it sometimes performs even worse than doing nothing in transfer learning.
>
> (1)  When training data are limited, the linear interpolation effect (which mixup pursues) tends to overfit, just like the commonly-known overfitting measure by the prediction accuracy. For example, if we have only two training examples, even by remembering infinite interpolated data between them, the learned model can hardly have such linear behaviors on unseen data pairs. In this sense, a direct application of mixup on a small training set will probably generalize bad (in terms of linear behaviors) on the test set, though it still benefits learning in general, as shown in our experiments of training from scratch (Figure 1b).
>
> (2) While the first reason is general to all scenarios with limited supervision, the second problem is particularly linked to the characteristic of transfer learning, where a powerful pre-trained model serves as the initialization. As figured out by existing studies like $L^2$-SP, DELTA and Co-tuning, a major challenge in transfer learning is the phenomenon called catastrophic forgetting. Unfortunately, the additional interpolated images generated by mixup drive the fine-tuned model farther from the starting point, which aggravates the loss of transferable knowledge in the pre-trained model. That’s why mixup sometimes performs worse compared against vanilla fine-tuning.
> Therefore, the reason of mixup’s ineffectiveness can be summarized as, limited mixed samples fail to learn generalizable linear behaviors, and aggravate catastrophic forgetting of pre-trained models.
>
> **A3.** SMILE solves the problem of transfer learning with mixup by sample-to-feature mixup. Specifically, sample-to-feature mixup effectively addresses the above two problems caused by mixup. First of all, SMILE incorporates deep target features and additional source labels as additional targets for mixup fitting. This makes the learned linear interpolations more generalizable, since these two kinds of features ($g_t$ and $z_t^{’}$) contain much richer information than using target labels only. Such an assumption has been validated in Table 6, where SMILE achieves the lower Label-IL than mixup on the test set, while mixup achieves lower Label-IL on the training set (i.e., the vanilla mixup overfits). Moreover, the source model is employed to provide *pseudo ground truth* for mixed features, which encourages the reuse of source knowledge.

---

### Review · Reviewer_eYuc · 2022-10-04

**Summary Of Contributions:**

This paper proposes a simple, yet seemingly effective technique for transfer learning based on mixup. The core idea is applying mixup to both sample-to-label (on both the source and target domains) and sample-to-feature maps. The authors validate the effectiveness of this technique in image classification (primarily for ResNet-50 models pretrained with ImageNet) and NLP benchmarks (fine-tuning BERT for the sentiment classification task SST-5). The paper includes a small ablation study showing the importance of both additional loss terms.

**Broader Impact Concerns:**

I did not identify any concerns on the ethical implications of this work.

**Requested Changes:**

In my opinion, the manuscript still needs a major editing effort in order for it to published. Authors might need to improve readability and fix the most obvious mistakes for the paper to be easy enough to comprehend.

## Language

In the following, I list some examples of sentences that seem incorrect:
* "Thus, **there needs to estimate the feature vector** for any sample in the target dataset before having the CNN trained."
* "Later, the work ... introduces new algorithms that prevent the regularization **from the hurts to transfer learning**, where ... **proposes** to truncate the tail spectrum of the batch of gradients while .. proposes to truncate the ill-posed .."
* "multi-tasking algorithms" -- not sure if the authors meant "multi-task".
* "In this way, **there might need a way** to make compromises between the features learned from both ..."
* "They find that, fine-tuning can be substantially" -- not sure about punctuation here.
* "Our ablation studies in Section 4 show that the simple combination of fine-tuning and mixup strategies **does not well in** transfer learning settings from both **linear behaviors preservation** and generalization performance aspects."
* "ten-crop ensemble" -- not sure what this is and there was no reference provided. Is this a misprint?

Examples of stylistic changes that would benefit the paper:
* Sentence "The selected best configurations are used for all configurations" contains the word "configurations" twice.
* "As for baseline methods, we use the recommended choices of hyper-parameters reported in **their papers**." -- perhaps replace "their papers" with "corresponding papers"?
* "offering **tons** of well-trained features", "**blessed by the power** of large source datasets" -- word "tons" and expression "blessed by the power" are in my opinion incompatible with the style of a scientific publication.
* "consider a more actual scenario" -- perhaps "relevant" or "practical", or another word instead of "actual"?

## Mathematical Notation

Here we outline some mistakes in mathematical notation found on page 5 and additional questions that need to be clarified in the text:
  * Authors introduce $\omega^k_s$ for the $k$-th iteration of the model. But $\omega_s$ is the source (presumably fixed) model. Should not it be $\omega^k_t$, the target model instead?
  * Underneath Equation (2) authors write $L^{\rm Reg} (\omega,\omega_t,\xi)$. Should it be $L^{\rm Reg}(\omega,\omega_s,\xi)$ instead?
  * $L^{\rm MXP}$ is mentioned, but is never defined explicitly. In my opinion, the expression for the final loss optimized by the authors has to be provided with _all_ of its terms.
  * The spacing around minus sign in Equation (4) is confusing.
  * Authors mention the fact that $\lambda$ is sampled from the beta distribution, it would be interesting to know a little bit more about the reasons for this choice without referring to another publication.
  * In Equation (4) we see both $g_t(\cdot;\omega_s)$ and $g_s(\cdot;\omega)$? Why are the $s$ and $t$ subscripts used on both $g$ and $\omega$? Why does $g_t$ (presumably target features) depend on $\omega_s$ (source weights) and $g_s$ (presumably source features) depends on $\omega$ (target weights)? This is either a mistake or needs a clarification. Right now this is simply confusing.
* Equation (6) misses a lot of closing brackets.
* $P_\delta$ is mentioned, but is not defined (is it the same distribution as $P_\lambda$?).
* "1e-4" in the text should be replaced with $10^{-4}$.

Please review the text thoroughly and fix all similar issues.

## Structure

There are Sections 4.0.1 and 4.0.2 that seemingly belong to Section 4.1 and should thus be Sections 4.1.1 and 4.1.2.


**Strengths And Weaknesses:**

## Strengths

* The proposed technique is simple, yet empirical results seem to support its effectiveness for transfer learning compared to some other recent methods. (Unfortunately, I do not know additional state-of-the-art techniques comparison with which could be important here.)
* Authors consider both the image and natural language domains (though with only one task).

## Weaknesses

* The overall level of manuscript preparedness is very low as if it was still a draft and not a finished paper. There are numerous errors and inaccuracies in the text from the language to mathematical notation and even the text structure (clearly wrong order of sections).
* Many explanations in the text are poorly articulated and together with clear mistakes in mathematical notation, they make reading and understanding this manuscript exceptionally difficult.
* The interpolation loss ${\rm IL}(f)$ in Equation 6 is not clearly explained.
   1. Firstly, when we speak of generalization, we typically consider one metric and analyze it for both training and test data distributions. For example, we can look at the model accuracy or the loss and compare their values on the training set and a hold-out test set to see whether overfitting occurs. Similar analysis could be carried out with $L^{\rm FE}$ and $L^{\rm FC}$. The difference of these losses on the training and test datasets would already characterize the degree of "overfitting" occurring on the training set and it is not entirely clear why we need another loss characterizing the linear structure on the test set.
   2. Secondly, while it is clear that ${\rm IL}(f)$ vanishes for linear functions $f$, one can also come up with other similar measures be that with only two samples (increasing the order / number of $\delta$ terms), or more than two samples. Was this form of ${\rm IL}(f)$ chosen because of its simplicity compared to other alternatives, or does it possess any other advantageous properties?

---

> ### Author Response · Authors · 2022-10-31
> **Response to Reviewer eYuc (1/2)**
>
> Thank you for your critical comments and useful suggestions. Based on your advices, we have revised the manuscript. Here we summarize your major concerns with respect to the following four Weaknesses and present our responses.
>
> **W1. The interpolation loss $\mathrm{IL}(f)$ in Equation (6) is not clearly explained.**
>
> We list the two questions w.r.t. $\mathrm{IL}(f)$ and respond to them one-by-one.
>
> **Q1. Why do we need the interpolation loss $\mathrm{IL}(f)$ and how does it reflect generalization?**
>
> A1. We are sorry that the explanation about $\mathrm{IL}(f)$ might be too brief to understand. Actually, $\mathrm{IL}(f)$ is not a metric that directly characterizes the generalization performance, but we use $\mathrm{IL}(f)$ to verify our assumption that SMILE is capable of improving the "linear interpolation" effects, as the goal of mixup is to make deep neural networks behave linearly [1]  (the linear behaviors are assumed to connect with generalization performance in [1]). We follow [1] to leverage linearity of DNNs as a key measure of generalization.
>
> We agree that over-fitting should be characterized by generalization gaps, such as the difference of top-1 accuracy between the training set and test set. The generalization advantages of our algorithms over baselines can be easily verified by test accuracy listed in Table 2, as the training accuracy easily achieves 100% in fine-tuning (greater testing accuracy means smaller generalization gap).  While improving the generalization is the final goal of an algorithm, the interpolation loss is introduced to illustrate the internal mechanism that leads to the improvements (as we follow [1] to assume the linear behaviors of a DNN connect to its generalizability). This is why we discuss the interpolation loss $\mathrm{IL}(f)$ in Section 5 (Analysis) rather than Section 4 (Experiments).
>
> To address your comments, we would additionally clarify that, the terminology “generalization” used in Section 5 particularly refers to the generalization of the linear interpolation effects (or mixup effects), rather than the common-sense meaning.
>
> *[1] mixup: Beyond Empirical Risk Minimization. International Conference on Learning Representations. 2018.*
>
> **Q2. Why do we chose the form of $\mathrm{IL}(f)$ as in Equation (6)?**
>
> A2. We chose this form mainly for consistency with the widely adopted form of the vanilla mixup. To be compatible with our sample-to-feature mixup, we define the function f to be any part of a neural network (not limited to be the entire network). Compared to the vanilla mixup, our Equation (6) is used to characterize the linear interpolation effects after training, while vanilla mixup leverages it as a learning objective beyond the empirical loss. So, it’s reasonable that vanilla mixup achieves the lowest Label-$\mathrm{IL}(f)$ as shown in Table 6. We use this form of interpolation loss also due to its simpleness and straightforwardness.
>
> Based on above clarifications, we make corresponding improvements in our revised manuscript as follows. (1) We explain the interpolation loss in more details about its purpose. (2) We point out the particular meaning of “generalization” in Section 5. (3) We explain the reason of choosing the form in Equation (6).
>
> **W2. Language problems.**
>
> We have checked throughout the manuscript and fixed those problems you pointed out and similar ones. Among those problems, we would like to further clarify two of them as follows.
>
> **Q1. "They find that, fine-tuning can be substantially" -- not sure about punctuation here.**
>
> A1. The usage of punctuation has been corrected according to English grammar.
>
> **Q2. ”ten-crop ensemble" -- not sure what this is and there was no reference provided. Is this a misprint?**
>
> A2. Ten-crop ensemble is a technique to improve the test accuracy. Specifically, for a test image, it generates ten random crops with random positions and sizes, and then aggregates the predictions of them to make a final prediction. This technique is sometimes used in transfer learning literature but not very common. We add the citation that also mentioned it.

---

> ### Author Response · Authors · 2022-10-31
> **Response to Reviewer eYuc (2/2)**
>
> **W3. Mathematical notation problems.**
>
> We have checked all mathematical notations and fixed similar issues. Further explanations are provided as the answers to the questions as follows.
>
> **Q1. Incorrect notations about the subscripts $s$ and $t$.**
>
> A1. In the original submission, we designed the notations based on a teacher-student framework, where $t$ refers to the teacher (source model), and $s$ refers to the student (target model). However, in the current version, we follow the custom to adopt the terminologies “source” and “target”, which is used more widely in the community of transfer learning. These problems have been fixed in the revised manuscript.
>
> **Q2. Why are the and subscripts used on both and $g$ and $w$?**
>
> A2. We use subscripts on $g$ to emphasize that the source and target feature extractor depend on different parameters, or they are different functions.
>
> **W4. The structure problems.**
>
> We have placed the subsection of “4.1 Image Classification” in a wrong position in the previous manuscript. Now it is fixed.

---

### Review · Reviewer_AUGU · 2022-10-09

**Summary Of Contributions:**

This work primarily focuses on making Mixup strategies work for deep transfer learning. While the motivation why Mixup should be used for transfer learning is not very clear, the results seem promising.  The proposed method efficiently leverages existing techniques like Mixup, Mean Teacher to improve performance in transfer learning. While the novelty is limited, it’s a simple solution that works in practice.

**Requested Changes:**

Experiments and Writing:
* The introduction section is confusing and doesn’t motivate “Why deep transfer learning would benefit from Mixup?”. I understand there are marginal gains, but what would be the primary advantage of using Mixup in Transfer Learning context? This needs to be motivated better.
* Experiments to show the effectiveness of the proposed Mixup strategy when used in combination with existing SoTA transfer learning methods like Co-Tuning and RegSL
* Details on the experimental setting like data splits, was a transfer learning benchmark used? If not, why?

Minor Issues:
* Error in Figure-1 legend: while the plot has dotted lines, the legend doesn’t have any dotted lines.
* Use of terms like source domain and target domain can be misleading and lead to questions and comparisons with domain adaptation; It might be better to use source dataset/target dataset instead of source domain and target domain.


**Strengths And Weaknesses:**

Strengths:
+ Proposes an empirically sound methodology for leveraging Mixup in the context of deep transfer learning.
+ Reasonable gains in terms of empirical performance.

Weaknesses:
- The motivation for why deep transfer learning would benefit from Mixup is not clearly discussed. In particular, the introduction section of the paper is a bit confusing and doesn’t set the proper context.
- For a more comprehensive comparison of existing methods, it would be interesting to see the results when the proposed Mixup methodology is added to more recent transfer learning approaches like Co-Tuning, RegSL considered in the paper.

Questions:
* Comparison with existing methods: Are all the existing methods re-trained on the data splits generated for the experiments, or was a standard transfer learning benchmark used?  Using benchmarks makes the comparison fair. For example, Guo et al., 2018, SpotTune [1], uses the Visual Decathlon benchmark. There are more recent benchmarks like Jiang et al. 2020 [2]

* “As the size of training data becomes smaller, our method yields more significant benefits,” -- As shown in the results, as the data size increases, the gains decrease, does this mean as the architecture is relying more on the pre-trained features instead of learning features specific to the target domain?

[1] Guo, Yunhui, Shi, Honghui, Kumar, Abhishek, Grauman, Kristen, Rosing, Tajana, and Rogerio Feris. "SpotTune: Transfer Learning through Adaptive Fine-tuning." arXiv, (2018). https://doi.org/10.48550/arXiv.1811.08737.
[2] Junguang Jiang, Baixu Chen, Bo Fu, Mingsheng Long, Transfer-Learning-library, Github repository, 2020, https://github.com/thuml/Transfer-Learning-Library

---

> ### Author Response · Authors · 2022-10-31
> **Response to Reviewer AUGU (1/2)**
>
> Thank you for your critical comments and constructive suggestions. Here, we summarize your major concerns/questions and present our responses/revisions in our manuscript.
>
> **W1. The motivation for why deep transfer learning would benefit from Mixup is not clearly discussed. In particular, the introduction section of the paper is a bit confusing and doesn’t set the proper context.**
>
> We are sorry if the manuscript makes you confused with our observations and motivations. Actually, the argument that deep transfer learning would benefit from Mixup is a straightforward conjecture that has been rarely investigated in previous works. Our preliminary experiments shown in Figure 1 demonstrated a simple combination of transfer learning and mixup that disproved this argument. Therefore, our main technical contribution is to propose a novel sample-to-feature Mixup that addresses the problems when using mixup in deep transfer learning for achieving better performance.
>
> Following your advices, we have improved the introduction section as follow. (1) In the added paragraph of Research Motivation, we explain why deep transfer learning would benefit from Mixup is a reasonable conjecture and figure out that, the vanilla Mixup actually tends to negatively affect deep transfer learning, when training samples are limited. (2) We analyze why this happens in the paragraph of Our Analyses. The following parts of the introduction section has clearly stated the technical challenges of this problem and our solutions, without heavy modifications.
>
> **W2. For a more comprehensive comparison of existing methods, it would be interesting to see the results when the proposed Mixup methodology is added to more recent transfer learning approaches like Co-Tuning, RegSL considered in the paper.**
>
> We agree that more results on such combinations should be interesting. Unfortunately, not all existing transfer learning algorithms are applicable for a simple combination with Mixup. For example, both Co-Tuning and RegSL rely on the original target labels in implementations. Co-tuning needs a conditional distribution between the source and target categories, i.e. $p(y_s|y_t)$. In case that each (mixed) sample is assigned a mixed label, the complexity to build the conditional probability distribution from source to target domains would grow explosively. It should be an interesting problem to extend the learned conditional distribution for supporting mixed labels, but goes beyond the scope of our work. Similarly, RegSL incorporates self label-correction and self label-reweighting strategies, which also depend on the original target labels and cannot directly adapt to mixed labels.
>
> **Q1. Comparison with existing methods: Are all the existing methods re-trained on the data splits generated for the experiments, or was a standard transfer learning benchmark used? Using benchmarks makes the comparison fair.**
>
> Yes, all the existing methods are re-trained on the data splits generated for the experiments. We partially follow important existing literature[1,2] to generate these data splits in the deep transfer learning community, and additionally evaluate on different source models, e.g. ResNet-50 trained on Places365 and EfficientNet-B4 trained on ImageNet. We would argue that there is not an absolutely standard benchmark among these existing literature, and therefore we use commonly adopted experimental settings.  We have a brief description of data splits in Section 4.1.1. In the revised manuscript, we add more details and explanations about the data splits.
>
> *[1] Catastrophic Forgetting Meets Negative Transfer: Batch Spectral Shrinkage for Safe Transfer Learning. (NeurIPS’19)*
>
> *[2] Co-Tuning for Transfer Learning. (NeurIPS’20)*
>
> **Q2. “As the size of training data becomes smaller, our method yields more significant benefits,” -- As shown in the results, as the data size increases, the gains decrease, does this mean as the architecture is relying more on the pre-trained features instead of learning features specific to the target domain?**
>
> Yes, we agree with your opinion. A general experience is that, when larger training data are available, improvements from additional algorithms (as well as pre-trained models) usually become less significant. In terms of deep transfer learning, when training samples are limited, the performance tends to be dominated by pre-trained features. This is because the target training data contain less domain-specific information due to its limited size.

---

> ### Author Response · Authors · 2022-10-31
> **Response to Reviewer AUGU (2/2)**
>
> **Q3. Use of terms like source domain and target domain can be misleading and lead to questions and comparisons with domain adaptation; It might be better to use source dataset/target dataset instead of source domain and target domain.**
>
> We agree that the term domain might lead to connections with domain adaptation, but domain indeed seems more suitable for the context of our work. The term domain usually contains to the concept of learned features (sometimes particularly the distribution) of a dataset. In this sense, fine-tuning algorithms, as well as ours, aim to transfer those general domain knowledge, despite the source and target tasks are different. To avoid misleading, we add a footnote clarifying the terminology in the first paragraph of the introduction in the revised manuscript.

---

### Author Response · Authors · 2022-10-31
**General Response**

Dear Editors and Reviewers,

Many thanks for your comments that improve our manuscript. We have revised the manuscript to address your comments.

To clarify the research motivation of our work, we have included additional discussions on the explaining why deep transfer learning would benefit from Mixup is a reasonable conjecture and figure out that, the vanilla Mixup actually tends to negatively affect deep transfer learning, when training samples are limited. We also analyze why this happens in the paragraph of Our Analyses. We have added the references that reviewers recommended and discuss these works in the context of our contributions. In terms of experiments, we included additional results required by reviewers to pilot the ideas of our work and confirm the advantages of our proposed methods. More analysis and elaboration have been done to shape our contributions better.

We fixed the language issues and make the mathematical notations consistent throughout the manuscript. Hope this work could receive your full consideration of acceptance for publication. Thank you!

Cheers,

Authors

---

### Decision · Action_Editors · 2022-12-11

**Recommendation:** Accept with minor revision

**Comment:**

This paper studies Sample-to-feature Mixup, an adapted Mixup technique for transfer learning, targeting on improving the data-efficiency of low target data regime. Four expert reviewers provided detailed and constructive comments for this paper; In correspondence, the authors provided a revised version with a large ratio of revised materials, which significantly strengthened the paper. The AE ensured that all reviewers took the revised version as well as the authors' response into consideration. In the final recommendation phase, two reviewers voted for acceptance but the other two reviewers voted for rejection. Two of them provided further information to justify their recommendations, which were highly appreciated.

AE justified all the material as well as the recommendation as a neutral referee, and reached the conclusion of "Revision". The main concern is that, this paper is an empirical study of a new transfer learning technique, but the current experimentation lacks deeper insights and background mechanism of mixup for transfer learning. In particular, why such a variant of Mixup with sample-to-feature strategy is useful for transfer learning--is it also useful for general-purpose supervised learning? While a theoretical understanding is definitely hard since Mixup itself is a data augmentation technique, a thorough empirical investigation is needed, as suggested by the reviewers. Also, since the mixup-related work is fruitful in our community, fine-tuning + some Mixup variants (e.g. Manifold Mixup, MixCut) shall be studied to make the evaluation more complete.

After confirmation with the EiC, AE would offer the authors with an opportunity for a revision. Authors shall be careful in addressing the major concerns.

**Audience:**

Yes, some individuals in TMLR's audience who are doing research in pre-training, fine-tuning and transfer learning would be interested in knowing the findings of this paper.

**Claims And Evidence:**

Not sufficiently. The claims are made at the abstract and introduction that sample-to-feature mixup is useful for transfer learning. Such a claim was studied empirically through experiments. While an empirical investigation is not unacceptable, the current experimentation is relatively lacking and more ablations are required for each design of the sample-to-feature mixup, e.g. the FC and FE terms in Eq. (5)--(6).

---

> ### Author Response · Authors · 2023-01-21
> **Addressing remained concerns and uploading camera-ready version**
>
> Dear Action Editors,
>
> Thank you for the decision of acceptance and your constructive suggestions. Following your advice, we have carefully revised our paper to address the concerns you mentioned, and reached a camery-ready version. Compared with the last version, our camery-ready paper made several improvements as follows.
> 1) We added more empirical investigation to explain the mechanism of our method. In particular, we figured out the importance of the source model in our framework by additional experimental analyses. The new contents are presented in Section 5.4.
> 2) We added show cases to demonstrated the effects of feature interpolation in Section 5.2.2 and Appendix B.
> 3) We evaluated Manifold Mixup and CutMix in our main experiment. Results show that those advanced stategies are highly competitive in standard benchmarks, while our method still performs better in low-data fine-tuning tasks.
> 4) We released our code, and provided the link in our camery-ready paper.
>
> Cheers,
>
> Authors